# Do Contextual and Demographic Factors Help Malaysian Nurses Prepare in Dealing with the COVID-19 Pandemic?

**DOI:** 10.3390/ijerph19095097

**Published:** 2022-04-22

**Authors:** Bee Seok Chua, Getrude Cosmas, Norkiah Arsat, Walton Wider

**Affiliations:** 1Faculty of Psychology and Education, University Malaysia Sabah, Kota Kinabalu 88400, Malaysia; chuabs@ums.edu.my (B.S.C.); getrudec@ums.edu.my (G.C.); 2Faculty of Medicine and Health Sciences, University Malaysia Sabah, Kota Kinabalu 88400, Malaysia; norkiah.arsat@ums.edu.my; 3Faculty of Business and Communication, INTI International University, Nilai 71800, Malaysia

**Keywords:** COVID-19, Malaysian nurses’ preparedness, age, working experience, actual major disaster event

## Abstract

As of 11 October 2020, Sabah comprised a quarter of all COVID19 cases reported in Malaysia since the start of the pandemic last January 2020. In this pandemic, nurses serve in vital roles to mitigate the effects of COVID-19. This study aims to explore the nurses’ preparedness in managing the COVID-19 outbreak situation in Sabah, Malaysia, examining the contextual (e.g., participated in an actual major disaster event) and demographic factors (e.g., age and working experience that may influence nurses’ preparedness for managing the COVID-19 pandemic). A total of 317 nurses in Sabah, Malaysia were involved in this study. The Emergency Preparedness Information Questionnaire (EPIQ) contained 41 items and was used to assess civilian nurses’ perceived familiarity with 9 competency dimensions of preparedness. The mean scores of nurses’ preparedness indicated a moderate level of nurses’ preparedness in managing the COVID-19 pandemic (mean ranging from 2.89 to 3.79). The results indicated that there was a significant difference between the nurses who actively participate in an actual major disaster event and who were not active in a total of preparedness and all dimensions (*t* = 2.79, df = 285, *p* = 0.006) (except in familiarity with decontamination and familiarity with special populations), across working experience (F(2,291) = 5.09, *p* = 0.007) (except familiarity with Incident Command System and role), and age among nurses ((F(3,290) = 2.68, *p* = 0.047)) (total of preparedness, familiarity with ethical issues in triage, epidemiology, and surveillance, and psychological issues). Overall, this study has made a significant theoretical contribution, as well as in clinical implications in the field of nursing practice, by addressing the impact of contextual and demographic factors on nurses’ preparedness in managing the COVID-19 outbreak situation. This study will help nurses to understand the skills, abilities, knowledge, and actions needed to respond, mitigate, and prepare for emergencies during pandemics and disasters.

## 1. Introduction

Nurses play a key role in the public health response to such crises, delivering direct patient care and reducing the risk of exposure to infectious disease [1]. According to the World Health Organization (WHO), nurses are the largest group of health professionals and the frontline of the health care system response to both epidemics and pandemics. Nurses have contact with patients and they are often directly exposed to this virus. Despite having a professional obligation to care for the community during a pandemic or epidemic, many nurses have concerns about their work and its impact on them personally [1]. The impact is such as psychological factors, feeling worried and anxious about being infected, and the possibility to transmit the virus to their family members. Anxiety related to the COVID-19 pandemic is prevalent in the nursing workforce, potentially affecting nurses’ well-being and work performance [2].

The Malaysian government is concerned about the burden and risks faced by nurses in taking care of COVID-19 patients. Therefore, the Malaysian government has provided RM600 million to the Health Ministry for hiring new medical staff, particularly nurses. This can reduce the work burden among nurses in treating the COVID-19 patient and in confronting the COVID-19 pandemic [3]. The WHO also recognizes the importance of nurses’ mental health and well-being in battling the pandemic. It is suggested that nurses need to be supported through psychological interventions to promote emotional release and improve their mental health [4].

Regarding nurses in Sabah, Malaysia, in dealing with COVID-19 patients, some of them were infected and have tested positive for COVID-19. This, as stated by the Malaysian Health Director-General Tan Sri Noor Hisham Abdullah, confirmed that two nurses posted in Queen Elizabeth II Hospital in Kota Kinabalu, Sabah, tested positive for COVID-19 (Emanuel, 2020). Additionally, healthcare workers at the intensive care unit (ICU) of Queen Elizabeth II Hospital in Kota Kinabalu, Sabah are struggling to cope with their workload after over half the nurses in the department were forced to self-quarantine. Malaysiakini reported that 40 of the 66 nurses operating the ICU wards were quarantined after one of their colleagues tested positive for COVID-19 [5].

Despite the negative effect caused by the pandemic, the health sector never closed down, and there was even greater demand to provide health care delivery services. Therefore, nurses are expected to be highly prepared towards fighting the disease [6]. Furthermore, nurses are also expected to be prepared to respond despite an emergency, and they should have the skills to oversee special needs patient, such as senior citizens, youngsters, and disabled persons [7]. However, past studies uncovered that the preparedness of nurses in facing the COVID-19 pandemic was not up to the ideal level [6,8,9,10]. In like manner, due to insufficient studies addressing the nurses’ preparedness in dealing with the COVID-19 pandemic [11], this study was conducted to examine the degree of preparedness of Malaysian nurses. It is anticipated that the findings in this study will help the Ministry of Health Malaysia to have a deeper understanding on the effect of contextual and demographic factors on nurses’ preparedness.

### 1.1. Background

Even before the COVID-19 pandemic, nurses played an active role in facilitating the community affected by major disasters such as earthquakes and hurricanes [12]. According to [13], disaster is defined as “a serious disturbance of the functioning of a community or a society causing huge human, material, financial or natural misfortunes that surpass the capacity of the affected local area or society to adapt utilizing its own resources”. However, Malaysia, unlike other countries, has not been struck by a huge scope disaster (e.g., volcanoes and tsunamis), on account of its topographical geology [14]. Rather, Malaysia regularly faces localized disasters, such as flood [15]. In 2015, Sabah was hit by an earthquake with a magnitude of 5.9, which caused 18 fatalities [16]. The serious seismic event caused the Sabah Health Department to establish the disaster management as a guide for management in terms of overall health aspects in any disaster incidents in Sabah [17]. Past studies had reported that involvement in the past disaster management is important to ensure that nurses are ready and prepared to face disaster when it strikes again in the future [14]. Therefore, it is important to determine the effect of nurses’ preparedness in managing the COVID-19 pandemic based on their participation in an actual major disaster event.

The unexpected coronavirus pandemic has caused great worries and anxiety among people around the globe about the accumulating cases and death rates [18]. These great worries also affected the frontline medical staff, such as nurses who deal directly with those who were affected [19]. These worries were also accompanied by the lack of COVID-19 equipment (e.g., personal protective equipment), facilities, and experienced nurses in dealing with a new outbreak virus and COVID-19 patients [20]. Furthermore, demographic characteristics such as age and working experience are also commonly discussed as important factors in increasing disaster preparedness among nurses [21,22]. Ref. [10] found that a higher level of preparedness in dealing with COVID-19 is associated with older and more experienced nurses in Jordanian hospitals. Similar findings were also reported in South Korea hospitals, where there is a positive correlation between disaster preparedness and age [21]. Experienced and long-servicing nurses might be more ready to face the COVID-19 patient compared to novice nurses, although they were equipped with knowledge during their training [23]. According to Al-Dossary et al., nurses need to develop a solid foundation of the disease process to play a greater role in disease control [24]. How nurses perceive and respond to COVID-19 is critical to expedite positive outcomes.

### 1.2. Objectives

The current study aims to explore the perceived familiarity and competency of nurses with response to preparedness in managing the COVID-19 outbreak situation in Sabah, Malaysia. This study also aims to examine the contextual (e.g., participated in an actual major disaster event) and demographic factors (e.g., age and working experience) that may influence nurses’ preparedness for managing the COVID-19 pandemic. The findings may help in creating awareness among the relevant authorities to understand more about nurses’ concerns and strengths in dealing with the COVID-19 pandemic.

## 2. Methods

### 2.1. Participants

There are 7066 registered nurses in public hospitals in Sabah, Malaysia [25]. The respondents of this study were registered nurses from the special practice, who served in the public hospitals in Sabah. To assess the minimum required sample size, ref. [26] suggested having a minimum of 250 samples for the results to be statistically valid and significant. The mean age of the nurses was 35.84 years (s.d = 8.55) and the mean of their working experience was 12.18 years (s.d = 9.01). There were 111 (or 35%) of the nurses who reported that they have actively participated in an actual major disaster event before, and 176 (55.5%) were not actively involved. Thirty of the nurses (9.5%) did not provide this information. Most of the nurses (79.8%) reported that they had never worked in a post-disaster shelter. In terms of the specialty practice area, 38.8% of the nurses were from the medical surgery specialty practice area, 20.8% of the nurses specialize in OB-Gyn, 12.6% of them were from the critical care practice area, and the rest were from specialty areas such as pediatric, operation room, and emergency services. Ref. [27] has gathered a very comprehensive data collection involving 3532 nurses throughout Sabah, Malaysia. Their respondents’ demographic profile shows a similar ratio to the current study in terms of age groups and working experience groups. Thus, this justified the unequal numbers of sample size with regard to age and working experience.

### 2.2. Ethical Considerations

Ethical consideration and approval to conduct the study in hospitals and district health offices were obtained from the Medical Research and Ethics Committee (MREC), and National Medical Research Register (NMRR) (ref. no: NMRR-20-1494-55626 [IIR]) of Malaysia, Ministry of Health (KKM) and the Medical Research Ethics Committee of the Faculty of Medicine and Health Sciences, Universiti Malaysia Sabah (Approval Code: JKEtika 3/20 (13)).

### 2.3. Data Collection

Participants in the current study included 317 nurses recruited from 4 hospitals in Sabah, Malaysia. The data of the nurses were collected using a self-administered survey method, and the questionnaire was distributed to the participants through the identified matron in the hospitals by using a convenience sampling technique. The hospital matrons were briefed on the administering of questionnaires. All involved nurses in the data collection were briefed on the purpose of the study and informed that their participation was completely voluntarily and confidential. In addition, they had the privilege to decline addressing any of the questions and, therefore, withdrawal was allowed. Completed surveys were stored in sealed envelopes to guarantee secrecy and were not accessible to anybody.

### 2.4. Instruments

A set of questionnaires contained demographic information (e.g., age, ethnicity, years of experience in the nursing, role, and specialty area) and the adapted Emergency Preparedness Information Questionnaire (EPIQ) by [28] was used to collect the data. The EPIQ consisted of 41 items to measure civilian nurses’ perceived familiarity with 9 competency dimensions of emergency preparedness: Familiarity with emergency preparedness terms and activities (e.g., “how to evaluate the effectiveness of your own actions during COVID-19 pandemic”); familiarity with the Incident Command System (ICS) (e.g., “assess and respond to site safety issues for self, co-workers, and affected people during COVID-19 pandemic”); familiarity with ethical issues in triage (e.g., “how to perform a rapid physical assessment of a patient of COVID-19”); familiarity with epidemiology and surveillance (e.g., “ability to identify the exacerbation of an underlying disease as a result of exposure to COVID-19 virus”); familiarity with decontamination (e.g., “the impact on the environment from the COVID-19 pandemic”); familiarity with communication/connectivity (e.g., “appropriate debriefing activities during COVID-19 pandemic”); familiarity with psychological issues (e.g., “appropriate psychological support for all parties involved in the COVID-19 pandemic”); familiarity with special populations (e.g., “the appropriate care of sensitive/vulnerable patient groups during the COVID-19 pandemic (i.e., aged, pregnant women, and the disabled)); and familiarity with accessing critical resources (e.g., “please provide an assessment of your overall familiarity with response activities/preparedness during COVID-19 pandemic”). For each item, respondents were requested to answer using a 5-point Likert scale (1 = Not Competent and 5 = Highly Competent). The EPIQ is considered to be the most reliable tool to identify perceived familiarity of emergency preparedness and disaster response core competencies among nurses [29].

The total score of the EPIQ is used as a measure of a total of nurses’ perceived competence in disaster preparedness. The EPIQ showed a high level of reliability in [7] study using a Malaysian Sample. The internal consistency for the total score on the EPIQ was Cronbach’s Alpha = 0.91 and, for the nine dimensions, the Cronbach’s Alpha coefficient ranged from 0.74 to 0.93 [7]. The EPIQ also showed an excellent level of reliability in the current study, with the Cronbach’s Alpha coefficient range from 0.829 to 0.942 for the nine dimensions and Cronbach’s Alpha coefficient = 0.97 for the total EPIQ.

### 2.5. Data Analysis

The normality of the data was first analyzed by alluding to skewness and kurtosis. Kim suggested the value of skewness < 2 and kurtosis < 7 to be absolute. Data collected were analyzed using Program IBM SPSS Statistic version 25.0. Statistical significance was defined as *p* < 0.05. Descriptive data were described using means, standard deviations, and ranges score of preparedness in managing the COVID-19 pandemic among the nurses in Sabah, Malaysia. Group differences in nurses’ preparedness were determined using independent *t*-test and One-Way Analysis of Variance. Table 1 indicates the descriptive statistics and normality results for the variables. Both skewness and kurtosis values were below 2, indicating that normality assumption was achieved for all the variables.

## 3. Results

### 3.1. The Level of Nurses’ Preparedness in Managing COVID-19 Pandemic

Among the nine dimensions, familiarity with accessing critical resources had the lowest mean score among the nurses, followed by familiarity with special populations and familiarity with epidemiology and surveillance, while the nurses reported the most familiarity with decontamination, communication/connectivity, and emergency preparedness terms and activities.

Participants’ mean item scores on our questionnaire ranged from 1.00 to 5.00, and the corresponding standard deviations ranged from 0.73 to 1.00. The mean scores and standard deviations for the nine dimensions were shown in Table 1. The mean scores indicated a moderate level of nurses’ preparedness in managing the COVID-19 outbreak situation in Sabah.

### 3.2. The Difference of Nurses’ Preparedness in Managing COVID-19 Pandemic Based on Their Participation in an Actual Major Disaster Event

The result of the *t*-test analysis revealed that the nurses who actively participated in an actual major emergency event were more prepared or competent in the total of preparedness in managing the COVID-19 pandemic. Specifically, results show that the nurses who participated actively in an actual major disaster had a significantly higher familiarity with emergency preparedness terms and activities, the Incident Command System (ICS) and role, ethical issues in triage, epidemiology and surveillance, communication/connectivity, psychological issues, special populations, and accessing critical resources than the nurses who were not actively involved in an actual major emergency event (refer to Table 2).

### 3.3. The Difference of Nurses’ Preparedness in Managing the COVID-19 Pandemic across Age Groups

The result of the one-way ANOVA revealed that the total of preparedness (F(3,290) = 2.68, *p* = 0.047), familiarity with ethical issues in triage (F(3,313) = 2.77, *p* = 0.042), familiarity with epidemiology and surveillance (F(3,311) = 4.22, *p* = 0.006), and familiarity with psychological issues (F(3,313) = 3.97, *p* = 0.008) was different across nurses’ age.

Post-hoc test using Tukey’s HSD indicated that the nurses in the age range of 31 to 35 years old perceived themselves to have higher competencies in total preparedness, familiarity with ethical issues in triage, familiarity with epidemiology and surveillance, and familiarity with psychological issues as compared to the nurses in the age above 40 years old. Additionally, the nurses in the age group below 30 years old also reported more familiarity with psychological issues than the nurses in the age above 40 years old (refer to Table 3).

### 3.4. The Difference of Nurses’ Preparedness in Managing the COVID-19 Pandemic across Nurses’ Working Experience

The result also showed that there were statistically significant differences between nurses across working experience in the total of preparedness (F(2,291) = 5.09, *p* = 0.007), and all the dimensions except the familiarity with the Incident Command System (ICS) and role (F(2,309) = 2.33, *p* = 0.099). The nurses with working experience of 8 to 15 years had significantly better preparedness in managing the COVID-19 pandemic and higher competencies in all these dimensions than the nurses with working experience above 15 years and the nurses with working experience of 7 years and below (in the familiarity of epidemiology and surveillance) (refer to Table 4).

## 4. Discussion

The fundamental purpose of this research is to examine the contextual and demographic factors that may influence nurses’ preparedness for managing the COVID-19 pandemic. Our results show a moderate level of preparedness in managing the COVID-19 pandemic among nurses in Sabah, Malaysia. Nurses who actively participate in an actual major disaster event showed a higher total of preparedness, specifically in familiarity with emergency preparedness terms and activities, the Incident Command System (ICS) and role, familiarity with ethical issues in triage, familiarity with epidemiology and surveillance, familiarity with communication/connectivity, familiarity with psychological issues, and familiarity with accessing critical resources. Meanwhile, for age, nurses in the age range of 31 to 35 years old have higher competencies in total preparedness, familiarity with ethical issues in triage, familiarity with epidemiology and surveillance, and familiarity with psychological issues compared to the other age group. In addition, our results also reported that across working experience, nurses with 8 to 15 years of working experience had better preparedness in managing the COVID-19 pandemic (except for familiarity with communication/connectivity).

In the present study, the mean scores indicated a moderate level of nurses’ preparedness in managing the COVID-19 outbreak situation in Sabah, Malaysia. Our findings are consistent with [6,8,9,10]. Among the nine dimensions, familiarity with accessing critical resources had the lowest mean score among the nurses, followed by familiarity with special populations and familiarity with epidemiology and surveillance, while the nurses reported the greatest familiarity with decontamination, familiarity with communication/connectivity, and familiarity with emergency preparedness terms and activities. Many studies that used EPIQ have frequently reported respondents as being most familiar with emergency preparedness terms [6]. However, our findings are somehow different, as reported by [29]. They reported that nurses in Saudi Arabia have a good preparedness in all theoretical dimensions of major incidents and disasters management, including emergency preparedness terms and activities, Incident Command Systems and their role in management, ethical issues in triage, epidemiology and surveillance, isolation and quarantine, decontamination, communication issues, psychological issues, management of special/vulnerable populations, and assessment of critical resources. What may be able to explain the difference between the findings of [29] and the current study is that the respondents of the previous study were only among nurses working in the emergency department. This indicates that all respondents of their study can be categorized as actively participating in actual major disaster events. The literature has consistently reported that nurses’ preparedness is strongly influenced by experience in managing disasters [22]. Therefore, the moderate level of preparedness in our findings is probably due to the fact that only 35% of nurses in our sample had prior disaster response experience. In addition, from the aspect of knowledge, ref. [29] found that most nurses had good knowledge (familiar to very familiar) in dimensions of signs and symptoms (different biological agents, and better for Anthrax), modes of transmission, antidotes, and adverse reactions, indicating a good preparedness in all theoretical dimensions due to most of their respondents having the tertiary level of nursing education. However, nurses seemed to be uncertain about their practical capabilities, skills, and evaluations of their own actions, including necessary first aid interventions such as ventilation and oxygen administration during a public health emergency [29]. Notably, in the present study, most nurses only had a nursing diploma qualification. However, further research needs to be conducted by further exploring the familiarity with accessing critical resources, familiarity with special populations, and familiarity with epidemiology and surveillance among the nurses. Interestingly, Baack et al. [30], on their samples with no exclusion criteria, suggested a somewhat low overall perceived competence to their respondent’s familiarity with disaster.

Notably, nurses play a leading role in preventing and responding to an outbreak, including COVID-19, throughout the healthcare sector in any country. The nurse is the front-line health service provider closest to the patient before another approaches the patient and is with the patient 24 h, no matter where the patient is and under any circumstances, whether facing complex illnesses that require hospitalization and even intensive critical care, such as COVID-19 [31]. Therefore, nurses’ familiarity, competency, and level of preparedness are critical in handling COVID-19 pandemics for them to be able to respond efficiently. This study is expected to help health service providers to better understand the readiness of nurses to provides care, in order to have a better workforce, training, and educational arrangement during emergencies of a pandemic nature.

The present study findings also indicated that there was a significant difference between the nurses who actively participated in an actual major disaster event and who were not active in a total of preparedness and all dimensions (except in familiarity with decontamination and familiarity with special populations), three categories of working experience (excepted in familiarity with the Incident Command System (ICS) and role). The result of the current study also indicated that there was a significant difference in the total of preparedness, familiarity with ethical issues in triage, familiarity with epidemiology and surveillance, and familiarity with psychological issues across four categories of age among nurses. This is contrary to the findings of [30], who reported that there was no significant relationship between age and nurses’ perceived preparedness. However, our study is consistent with [10,21], in which disaster preparedness among nurses is influenced by age.

The result also showed there was a significant difference between nurses across working experience in the total of preparedness and all the dimensions, except in familiarity with the Incident Command System (ICS) and role. Nurses with working experience of 8 to 15 years had significantly better preparedness in managing the COVID-19 pandemic and higher competencies in all these dimensions than the nurses with working experience above 15 years and the nurses with working experience of 7 years and below (in the familiarity of epidemiology and surveillance). However, nurses with working experience of 7 years and below had better familiarity with special populations than the nurses with working experience above 15 years. The nurses in the age range of 31 to 35 years old reported higher competencies in total preparedness, familiarity with ethical issues in triage, epidemiology, surveillance, and psychological issues compared to the nurses in the older age range (above 40 years old). Furthermore, the younger nurses (below 30 years old) also reported more competence in psychological issues than the older age nurses. In [30], a study that involved the nurses with an average of 42 years of age and 15 years of experience, they found that previous participation in a major disaster and experience in a post-disaster shelter were correlated with the total EPIQ score and significantly predicted the nurses’ perceived competence in disaster preparedness. Most nurses are not confident in their ability to respond to major disasters events. Confident nurses are more likely to have actual prior experience in disasters or shelters. Behavioral self-regulation (motivation) is a significant predictor of a nurse’s perceived competence to manage a disaster only if the nurse is willing to bear the risk of engaging in a disaster situation [30].

Meanwhile, emergency preparedness among nurses in China is at a low level, which is contrary to the results of the study of [29], and the current study suggests that training should be undertaken to improve the level of preparedness and better-quality response, as nurses with protection training and working experience in SARS tend to have higher levels of emergency preparedness [9]. As suggested earlier by [30], a positive correlation between previous experience and a higher score on training or participation in the actual event may increase nurses’ perceived preparedness and engender actual abilities in preparedness.

### Implication for Clinical Practice

The implication of this study on clinical practice is that there is a difference between nurses who are actively involved and not actively involved in actual disaster events, where nurses who are actively involved showed high level of readiness in all dimensions of theory, especially for respondents who have a level of nursing education at the tertiary level. Nurses with tertiary education are usually involved with management duties and responsibilities in nursing, therefore, although they show a higher level of readiness in theory, they are unsure about their practical ability, skills, and assessment of their own actions, including in first aid such as ventilation and oxygen delivery during public health emergencies. Since nurses with between 8 and 15 years of experience are young and productive nurses, they had a much better readiness in managing the COVID-19 pandemic and higher efficiency in all these dimensions compared to nurses with working experience above 15 years and nurses with working experience 7 years or less. This suggests that less experienced nurses are novice nurses and those with more than 15 years of experience are senior nurses who are in the category of nurses at an older age (above 40 years), who may have limitations in terms of physical and emotional ability in readiness to manage COVID-19 type pandemics. This was evidenced with nurses aged 31 to 35 years, reporting higher efficiencies in total readiness, familiarity with ethical issues in triage, epidemiology, supervision, and psychological issues compared to nurses at older ages (above 40 years).

Therefore, nursing management and policy makers in nursing education and services in Malaysia need to accelerate the process of standardizing the level of education and recruitment requirements for graduate nurses, so that the gap in academic qualifications between graduate nurses and nurses with only diploma qualifications can be minimized. The results of previous studies and current studies have found that graduates with a degree level of education have higher knowledge than nurses who only have a diploma. This affected their readiness in managing the COVID-19 outbreak. Further studies need to be conducted on nurses based on their background of nursing specialization, such as emergency nursing specialization and others. Meanwhile, from the aspect of experience and age of nurses, the nursing management needs to outline the conditions of nurse placement based on experience and appropriate age for nurses who will be placed in the COVID-19 pandemic management division to ensure nurses can function efficiently and effectively while managing disaster situations.

## 5. Conclusions

The present study has provided original investigation on the nurses’ preparedness in managing the COVID-19 outbreak situation in Sabah, Malaysia. Nonetheless, this study is constrained by some limitations. First, the surveys were circulated through the matron, who is assigned to the public hospitals for data collection. However, it is beyond the control of the researchers to avoid disturbance to the nurses on duty. Thus, confidentiality during the course of data collection was somewhat compromised. Second, the respondents were special practice nurses from four public hospitals in Sabah, Malaysia. Consequently, differences in nurses’ preparedness in managing COVID-19 may be found differently in other discipline of nurses. In order to generalize the results, future studies should examine nurses’ preparedness according to the discipline nurses are assigned to, such as nurses working in public hospitals and nurses working in health care services, as various settings may have different preparedness in managing the COVID-19 outbreak situation.

## Figures and Tables

**Table 1 ijerph-19-05097-t001:** The means, standard deviations, and ranges scores of preparedness in managing COVID-19 pandemic among the nurses (*n* = 317).

No	Dimension	Mean	Std. Deviation	Skewness	Kurtosis
1	Mean Score Term	3.34	0.73	−0.277	−0.304
2	MeanScore_ICS	3.28	0.83	−0.393	−0.376
3	MeanScore_ethical	3.20	0.89	−0.372	−0.666
4	MeanScore_epidemiology	3.16	0.94	−0.273	−0.544
5	MeanScore_Decontamination	3.79	0.87	−0.578	−0.038
6	MeanScore_Communication	3.42	0.89	−0.463	−0.293
7	MeanScore_Psychological	3.20	0.92	−0.135	−0.461
8	MeanScore_Resources	2.89	0.89	0.137	−0.714
9	MeanScore_Population	3.06	1.00	0.023	0.273

**Table 2 ijerph-19-05097-t002:** The *t*-test of nurses’ preparedness and its sub-dimensions in managing the COVID-19 pandemic based on their participation in an actual major disaster event.

No	Nurses Preparedness	Participated Actively in Actual Major Disaster Event	N	Mean	SD	T	*p*-Value
	Total of Preparedness	Yes	103	142.50	27.69	2.79	0.006 *
		No	168	132.54	29.06		
1.	Familiarity with emergency preparedness terms and activities	Yes	106	20.91	4.47	2.82	0.005 *
		No	171	19.43	4.12		
2.	Familiarity with the Incident Command System (ICS) and role	Yes	107	27.59	5.86	1.97	0.050 *
		No	176	26.06	6.64		
3.	Familiarity with ethical issues in triage	Yes	111	13.63	3.54	2.86	0.050 *
		No	176	12.42	3.46		
4.	Familiarity with epidemiology and surveillance	Yes	111	13.35	3.76	2.07	0.039 *
		No	175	12.43	3.57		
5.	Familiarity with decontamination	Yes	111	11.69	2.43	1.32	0.189
		No	176	11.27	2.76		
6.	Familiarity with communication/connectivity	Yes	109	25.63	5.73	2.88	0.004 *
		No	173	23.51	6.22		
7.	Familiarity with Psychological issues	Yes	111	13.68	3.23	2.49	0.013 *
		No	176	12.59	3.85		
8.	Familiarity with special populations	Yes	111	6.48	1.85	1.81	0.072
		No	176	6.04	2.09		
9.	Familiarity with accessing critical resources	Yes	111	9.48	2.39	3.93	0.000 *
		No	176	8.28	2.59		

Note. * The mean difference is significant at *p* < 0.05.

**Table 3 ijerph-19-05097-t003:** Analysis of variance of the nurses’ preparedness in managing COVID-19 pandemic across four categories of age.

No.	Nurses’ Preparedness	Working Experience	N	Mean	Std. Deviation	F	*p*-Value
	Total of Preparedness	30 years old and below	92	137.73	24.75	2.68	0.047 *
		31 to 35 years old	68	140.79	28.89		
		36 to 40 years old	49	136.86	29.93		
		above 40 years old	85	128.34	32.96		
1.	Familiarity with emergency preparedness terms and activities	30 years old and below	92	20.30	4.14	2.18	0.090
		31 to 35 years old	68	20.84	4.23		
2.		36 to 40 years old	50	20.14	3.89		
		above 40 years old	96	19.18	4.72		
3.	Familiarity with the Incident Command System (ICS) and role	30 years old and below	93	26.59	5.60	1.48	0.221
		31 to 35 years old	69	26.65	6.28		
		36 to 40 years old	52	27.13	6.65		
		above 40 years old	98	25.09	7.64		
4.	Familiarity with ethical issues in triage	30 years old and below	94	13.03	3.25	2.77	0.042 *
		31 to 35 years old	69	13.42	3.53		
		36 to 40 years old	53	13.04	3.19		
		above 40 years old	101	11.97	3.94		
5.	Familiarity with epidemiology and surveillance	30 years old and below	94	13.14	3.18	4.22	0.006 *
		31 to 35 years old	69	13.46	3.58		
		36 to 40 years old	53	12.45	3.65		
		above 40 years old	99	11.63	4.24		
6.	Familiarity with decontamination	30 years old and below	94	11.62	2.41	1.87	0.135
		31 to 35 years old	69	11.75	2.65		
		36 to 40 years old	53	11.28	2.71		
		above 40 years old	101	10.91	2.65		
7.	Familiarity with communication/connectivity	30 years old and below	94	24.63	4.87	0.77	0.510
		31 to 35 years old	69	24.59	6.57		
		36 to 40 years old	52	24.06	6.53		
		above 40 years old	93	23.40	6.78		
8.	Familiarity with psychological issues	30 years old and below	94	13.20	3.12	3.97	0.008 *
		31 to 35 years old	69	13.65	3.86		
		36 to 40 years old	53	12.87	3.46		
		above 40 years old	101	11.84	4.01		
9.	Familiarity with special populations	30 years old and below	94	6.28	1.73	2.41	0.067
		31 to 35 years old	69	6.52	2.22		
		36 to 40 years old	53	6.06	1.92		
		above 40 years old	101	5.73	2.10		
10.	Familiarity with accessing critical resources	30 years old and below	94	8.78	2.54	2.10	0.100
		31 to 35 years old	69	9.22	2.55		
		36 to 40 years old	53	8.74	2.71		
		above 40 years old	101	8.20	2.80		

Note. * The mean difference is significant at *p* < 0.05.

**Table 4 ijerph-19-05097-t004:** Analysis of variance of the nurses’ preparedness in managing COVID-19 pandemic across three categories of working experience.

No.	Nurses’ Preparedness	Working Experience	N	Mean	Std. Deviation	F	*p*-Value
	Total of Preparedness	7 years and below	114	134.90	25.61	5.09	0.007 *
		8 to 15 years	86	143.17	29.01		
		Above 15 years	94	129.45	32.55		
1.	Familiarity with emergency preparedness terms and activities	7 years and below	114	20.20	4.22	3.18	0.043 *
		8 to 15 years	88	20.78	4.06		
		Above 15 years	104	19.24	4.59		
2.	Familiarity with the Incident Command System (ICS) and role	7 years and below	114	26.08	5.76	2.33	0.099
		8 to 15 years	91	27.40	6.39		
		Above 15 years	107	25.38	7.55		
3.	Familiarity with ethical issues in triage	7 years and below	114	12.75	3.34	6.82	0.001 *
		8 to 15 years	93	13.78	3.15		
		Above 15 years	110	11.96	3.92		
4.	Familiarity with epidemiology and surveillance	7 years and below	114	12.90	3.29	5.47	0.005 *
		8 to 15 years	93	13.34	3.61		
		Above 15 years	108	11.69	4.18		
5.	Familiarity with decontamination	7 years and below	114	11.45	2.37	3.13	0.045 *
		8 to 15 years	93	11.81	2.69		
		Above 15 years	110	10.91	2.70		
6.	Familiarity with communication/connectivity	7 years and below	114	23.93	5.12	2.20	0.113
		8 to 15 years	92	25.23	6.61		
		Above 15 years	102	23.43	6.69		
7.	Familiarity with psychological issues	7 years and below	114	12.87	3.14	5.98	0.003 *
		8 to 15 years	93	13.73	3.86		
		Above 15 years	110	11.96	3.92		
8.	Familiarity with special populations	7 years and below	114	5.98	1.84	5.71	0.004 *
		8 to 15 years	93	6.69	2.04		
		Above 15 years	110	5.78	2.06		
9.	Familiarity with accessing critical resources	7 years and below	114	8.74	2.58	3.37	0.036 *
		8 to 15 years	93	9.17	2.57		
		Above 15 years	110	8.21	2.79		

Note. * The mean difference is significant at *p* < 0.05.

## Data Availability

Data can be made available upon reasonable request.

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
