# Peer review of "Do Contextual and Demographic Factors Help Malaysian Nurses Prepare in Dealing with the COVID-19 Pandemic?"

_ijerph, 2022, doi:10.3390/ijerph19095097_

Round 1

Reviewer 1 Report

Introduction
The introduction should include international studies that support what has been stated in this study. In addition, an adequate justification of the study is necessary. Why is this study important?  It has been overly contextualised and no international studies have been presented to support the importance of the study.
Methodology.
Indicate the sample calculation.
Ethical considerations regarding informed consent, anonymity of participants and confidentiality of data have not been described.
What was the procedure for data collection? It should be stated.
The data analysis should specify what tests were performed in the data analysis Were the variables tested for normality?
Line 121-122. Should be in the data analysis section.
Line 132-139.  Should be before table 1.
Line 143-149: Specify exactly where the significant differences are found according to the table described.
What limitations does this study actually have?
What implications does this study have for clinical practice?

Reviewer 2 Report

Thank you very much for giving me the opportunity of reviewing this paper regarding the nurses’ preparedness in dealing with the pandemic. Nurses are essential health professionals during the COVID- 19 pandemic, research to improve their work performance are needed.

Title

Please, reconsider the terms Contextual Factors and Past Experiences, as the study analyses age and working experience.

Abstract

Please, add some numerical results.

There should be a conclusion obtained from the results, not only an implication for practice.

Keywords

Please reconsider age, working experience and actual major disaster event.

Introduction

In general, the introduction lacks references, previous evidence on which to support the arguments that are exposed. There are only 6 references throughout the introduction, the covid-19 pandemic is being widely studied, as well as the role of nursing in this pandemic. I recommend that you support the introduction with more references.

Lines from 34 to 41 need references that support the information provided.

Lines 42: Please mention the author’s name, like according to Al-Dossary et al. [1].

The background of the subject (nurses’ preparedness) needs further development. You should describe it with more details, showing a deep and wide picture of the state of the art. Please, include antecedents and consequences of preparedness.

Methods

Please, expand the data collection procedure and how the questionnaire was disseminated.

Please describe with more details the The Emergency Preparedness Information Questionnaire. How is each item assessed, with a Likert scale? What are the maximum and minimum scores? What are the cut-off points? How is it interpreted?

Include an ethical section that include all the ethical issues such as informed consent, data protection, voluntary participation…

Results

Please, explain how the age groups were stated (table 4). They do not seem balanced, for example, the groups below 30 years and above 40 years look like much bigger than the others. How did you stablish the limits for each group?

Please, do the same for the groups regarding the years of experience (table 6).

Please consider joining tables 3 and table 4 in one table.

Please consider joining tables 5 and table 6 in one table.

Discussion

Present a brief report of the results at the beginning of the discussion (move lines 218-224)

Line 217: I don’t understand the usefulness of reference 10. This line stated the purpose of this study, it does not need to be supported.

The results obtained in this study are only compared with two previous studies (references 11 and 12). This is very poor and have to be improved. You need to compare your results with previous similar studies and provide evidence of your arguments that could explain similarities or differences.

Please include the limitations of your study.

Your recommendations for practice (lines 296-302) should not be a summary of the results. Please, state what would you suggest as improvement for practice in the light of your results.

Reviewer 3 Report

  1. In introduction part, the subtitle is better to insert for the first part. For example, 1.1 Backgrounds, 1.2 Objectives.
  2. You said ‘in terms of preparedness in managing the COVID-19 situation’ in objectives part. So how did you ask it for nurses in this study. You should be mentioned COVID-19 in your survey, so, please show the explanation for the participants.
  3. What is an actual major disaster event? What is your definition for that? You have to define an actual major disaster event.
  4. In objectives, because you said the purpose is to understand the factors contributes to nurses’ preparedness and readiness to face COVID-19, the data should be analyzed with regression. However, you just compared the mean of EPIQ according to demographic variables.
  5. Table 1 should be simplified. No need N. “N=317” should be shown just one time in the upper right side of table. The scale is 5-likert, So, Minimum and maximum was not needed to show. Delete it.
  6. The number of each dimension of EPIQ should be shown in the table.
  7. Table 1 and Table 2 are better to merge.
  8. In Table 2 and 3, Sig. should be changed to the p. p is significance probability.
  9. Table 3 and 4 also should be merged. Sum of squares, df, means square are not needed in table.
  10. Significance probability p and F should be shown in Table 4.
  11. Table 5 and 6 also should be merged. Sum of squares, df, means square are not needed in table.
  12. Actually, Table 1,2,3,4,5 and table 6 can be shown just in one table. The variables including participated activity, age, and working experience are left column vertically, and each dimension of preparedness are right horizontal line, and M±SD, t or F, and then p

Round 2

Reviewer 1 Report

The authors have made the indicated modifications, therefore the manuscript can be accepted.

Thank you very much

Author Response

We are grateful for your consideration of this manuscript, and we also very much appreciate your suggestions, which have been very helpful in improving the manuscript.

Reviewer 3 Report

Thank you for your efforts. I think you did your best for upgrade it. 

You did revised according to reviewers' comments including me. 

Just one point, Background as the subtitle of 1.1 is better than literature review. But, it's my opinion, so it's up to you. 

Moreover, English typo or spells should be checked one more time.   

Good Luck. 

Author Response

- 1.1 has been changed to Background as suggested.

- We are grateful for your consideration of this manuscript, and we also very much appreciate your suggestions, which have been very helpful in improving the manuscript. Thank you.